# Evaluation of Agronomic Characteristics, Disease Incidence, Yield Performance, and Aflatoxin Accumulation among Six Peanut Varieties (*Arachis hypogea* L.) Grown in Kenya

**DOI:** 10.3390/toxins15020111

**Published:** 2023-01-28

**Authors:** Loise Njoki, Sheila Okoth, Peter Wachira, Abigael Ouko, James Mwololo, Margherita Rizzu, Safa Oufensou, Truphosa Amakhobe

**Affiliations:** 1Department of Biology, University of Nairobi, Nairobi P.O. Box 30197-00100, Kenya; 2International Crops Research Institute for the Semi-Arid Tropics (ICRISAT), Lilongew P.O. Box 1096, Malawi; 3Dipartimento di Agraria, Italy Nucleo di Ricerca sulla Desertificazione, NRD, University of Sassari, Viale Italia 39/A, 07100 Sassari, Italy

**Keywords:** peanut diseases, aflatoxins, *Arachis hypogea* L., agronomic characteristics, yield, biomass, UHPLC-FLD

## Abstract

Diseases contribute to attainment of less than 50% of the local groundnut potential yield in Kenya. This study aimed to evaluate the agronomic characteristics (flowering and germination), disease incidence, yield performance (biomass, harvest index, 100-pod, 100-seed, and total pod weight), and aflatoxin accumulation in six peanut varieties. A field experiment was conducted using four newly improved peanut varieties: CG9, CG7, CG12, and ICGV-SM 90704 (Nsinjiro), and two locally used varieties: Homabay local (control) and 12991, and in a randomized complete block design with three replications. The disease identification followed the International Crop Research Institute for the Semi-Arid Tropics (ICRISAT) rating scale and further isolation of fungal contaminants was conducted by a direct plating technique using potato dextrose agar. The aflatoxin levels in the peanuts were determined after harvesting using the ultrahigh performance liquid chromatography and fluorescence detection (UHPLC-FLD) technique. ICGV-SM 90704 showed the least average disease incidence of 1.31 ± 1.75%, (*P* < 0.05); the lowest total aflatoxin levels (1.82 ± 1.41 μg kg^−1^) with a range 0.00–0.85 μg kg^−1^ for total aflatoxins and a range 0.00–1.24 μg kg^−1^ for Aflatoxin B_1_. The locally used varieties (12991 and the control) revealed the highest disease incidence (5.41 ± 8.31% and 7.41 ± 1.88%), respectively. ICGV-SM 90704 was the best performing among all the six varieties with an average total pod weight (9.22 ± 1.19 kg), 100-pod weight (262.93 ± 10.8 g), and biomass of (27.21 ± 5.05 kg) per row. The 12991 variety and the control showed the least total pod weight (1.60 ± 0.28 and 1.50 ± 1.11 kg, respectively) (*P* = 0.0001). The newly improved varieties showed lower disease rates, low levels of aflatoxins, and higher yields than the locally used varieties.

## 1. Introduction

Peanut (*Arachis hypogea* L.) is a major cash crop widely grown in the global tropical and subtropical regions, used as food, oil, fodder, and organic fertilizer [1]. Peanuts contain 48–50% oil, 26–28% protein, and are a rich source of dietary fiber, minerals (Ca, P, Mg, Zn, and Fe), and vitamins (E, K, and B complex) [1,2]. The shells are used as fuel and animal feed, cattle litter, and filler in the feed and fertilizer industry [2]. The biomass acts as animal fodder and organic fertilizer. Since peanuts are leguminous, they add nitrogen and organic matter to soil [1]. Africa and Asia produce 91% of the world’s total groundnut [3]. In Kenya, peanuts are potentially sustainable crops majorly grown in the western and Nyanza regions during the short and long rains [3]. The region uses the crop as a staple crop after maize due to its ability to grow during dry seasons and withstand the climatic conditions. The most crop growers are small-scale farmers who struggle in meeting the market demands of the crop due to biotic and abiotic stress factors, which reduce the yield and quality.

Among several biotic factors compromising peanut production, fungal diseases have the most significant adverse effects on yield, agronomic characteristics (flowering and germination), and quality [4]. The interaction between biotic factors and varieties significantly affect disease incidence and severity, which interfere with peanut yield [5]. Fungal diseases affecting foliar parts of the crop are the most common, such as rust caused by *Puccinia arachidis* and leaf spots (*Cercospora arachidis*), with fewer cases of soilborne diseases such as *Aspergillus* crown rot and southern stem rot [4]. The diseases arise at different stages from germination, flowering, and maturity. Storage and processing of the crop predispose the peanuts to aflatoxin contamination [3]. A study revealed that the most common diseases of peanuts are groundnut rosette and early and late leaf spot (ELS and LLS) with incidences of 84.6%, 64.4%, and 49.3%, respectively [6]. Another study [7] further indicated that rust and LLS were the two most widely distributed and economically important foliar diseases of groundnut causing severe damage to the crop.

These foliar diseases are commonly present where groundnut is grown but their incidence and severity differ between localities and seasons [7]. Each disease alone could result in substantial yield loss, but when the diseases occur together the losses are massive [7]. The foliar diseases do not only reduce the yield but also have negative effects on seed quality and alter the quality and quantity of plant biomass [7]. A qualitative study revealed that infestation by rust and LLS in peanuts caused a 35% loss in yield, and fungicides reduced the disease effects by 1.01% compared to the non-treated fields [8]. These foliar diseases, mainly rust and LLS, caused severe defoliation and altered pod maturity.

Peanuts are exposed to contamination with toxic fungal metabolites called aflatoxins [3,9]. Aflatoxins are produced by a number of *Aspergillus* species including the agriculturally important *A. flavus*, *A. parasiticus*, and *A. nomius* [3]. The metabolites thrive in tropical and subtropical areas due to high temperatures and humidity. Peanut pods and seeds stored under high temperatures and humidity, or predisposed to diseases and pests, show a higher risk of aflatoxin contamination. The toxic and carcinogenic nature of the mycotoxins to humans and livestock raise concerns since they contaminate major food crops such as peanuts [9]. In Kenya, the minimum threshold for total aflatoxin levels in peanuts is 10 ug kg^−1^ [9,10]. Rates above the minimum threshold increase risk factors to consumers, yet high levels of the metabolite are recorded for staple foods such peanut and maize [10]. Determining the levels of aflatoxins in peanuts and their influencing factors helps in raising awareness and developing ways to mitigate the contamination rates. Evaluating the exposure of peanut varieties to diseases, aflatoxin contamination, agronomic characteristics, and yield performance offers a sustainable platform for adopting high-quality groundnuts for food safety and economic stability. The selected study region, Nyakach, has a majority of households dependent on agriculture and livestock rearing as their livelihood. Ecologically, the region spreads across two main agroecological zones (AEZs): LM3 and LM4 [11,12]. The semiarid region receives unreliable rainfall that accounts for an increased risk of diseases and mycotoxin contamination in staple crops such as peanut.

The objective of this study was to determine the agronomic characteristics, disease incidence, yield performance, and aflatoxin accumulation in selected peanut varieties, in Nyakach, Western Kenya, in order to identify the best performing varieties in terms of yield, disease susceptibility, and aflatoxin contamination.

## 2. Results

### 2.1. Agronomic Performance of the Six Varieties

#### 2.1.1. Germination

The variety CG7 showed the highest germination rate of 60.75 ± 2.48% after 15 days from the day of planting, followed by ICGV-SM 90704 at a rate of 54.53 ± 4.88%. The least rate of germination (11.63 ± 0.99%) was recorded in 12991 (Figure 1). The ANOVA and Tukey HSD test revealed that at 5% level of confidence, there were significant differences between the germination rates of the six peanut varieties (*P* = 0.000) (Figure 1).

#### 2.1.2. Flowering

The variety CG12 showed the least days to 50% flowering (40.94 ± 0.32), followed by CG9 (40.97 ± 0.23). The variety 12991 and the control (Homabay local) had the most days to 50% flowering (44.23 ± 0.79 and 44.40 ± 0.40, respectively). CG9 had the least days to 75% flowering (42.79 ± 0.23) followed by CG12 at a rate of (42.87 ± 0.32). The locally used sample (12991) had the most days to 75% flowering (49.51 ± 1.09) (Figure 1). The ANOVA and Tukey HSD test revealed that at the 5% level of confidence, there was a significant difference between the flowering rates at 50% and 75% of the six peanut varieties (*P* = 0.000) (Figure 1).

### 2.2. Disease Incidence among the Six Varieties

The most prevalent peanut disease among the five varieties was the groundnut rosette virus (4.01 ± 0.44%), followed by early leaf spot (2.79 ± 0.20%) and LLS (2.34 ± 0.24%), which are foliar diseases. The least prevalent disease was *Aspergillus* crown rot (ACR) (0.75 ± 0.21%), which is a stem and root disease. The Tukey HSD test revealed that the prevalence of the diseases among the five varieties was significantly different (*P* < 0.05) (Table 1).

The peanut variety with the highest disease rate was the control (Homabay local) (7.41 ± 1.88), followed by the locally used variety, 12991 (5.41 ± 8.31%), and the least diseased was ICGV-SM 90704 (1.31 ± 1.75%). The ANOVA and Tukey HSD test revealed that at the 5% level of significance, the disease incidence (per seed planted) of the different peanut varieties was significantly different (*P* = 0.000) (Table 1).

### 2.3. Yield Performance of the Six Varieties

#### 2.3.1. Total Pod Weight

The peanut variety with the highest total pod weight was ICGV-SM 90704 (9.22 ± 1.19 kg), followed by CG7 (8.86 ± 1.45 kg) and CG12 (7.29 ± 0.56 kg). Variety 12991 and the control samples had the lowest total pod weight (1.60 ± 0.28 and 1.50 ± 1.11 kg, respectively). The ANOVA and Tukey HSD test at 5% significance revealed that there was a significant difference in pod weight among the peanut varieties (Table 2).

#### 2.3.2. Biomass

Variety ICGV-SM 90704 had the highest average biomass per row (27.21 ± 5.05 kg), followed by CG7 (25.06 ± 2.4 kg) and CG12 (17.67 ± 2.63 kg). The control (Homabay local) sample and 12991 had the least biomass of 6.60 ± 1.20 and 5.45 ± 0.54 kg, respectively. The Tukey HSD test at 5% significance revealed a significant difference in biomass among the six varieties (Table 2).

#### 2.3.3. Harvest Index

Variety ICGV-SM 90704 had the highest harvest index per row (29.21 ± 0.72%), followed by CG7 (26.12 ± 0.91%) and CG12 (25.31 ± 1.22%). The control (Homabay local) sample and 12991 had the least harvest index of 18.52 ± 1.44% and 22.00 ± 0.16%, respectively (Table 2).

#### 2.3.4. 100-Pod Weight

The variety with the highest 100-pod weight was ICGV-SM 90704 (262.93 ± 10.8 g), followed by CG7 (260.84 ± 4.6 g) and CG9 (259.37 ± 6.4 g). The two varieties with the least 100-pod weight were 12991 (181.46 ± 6.44 g) and the control (Homabay local) (150.80 ± 11.4 g). The Tukey HSD test revealed a significant difference in the 100-pod weight between the six peanut varieties *P* = 0.000 (Table 3).

#### 2.3.5. 100-Seed Weight

The variety with the highest 100-seed weight was CG9 (118.55 ± 6.03 g), followed by ICGV-SM 90704 (115.71 ± 15.24 g) and CG7 (111.43 ± 2.12 g). CG12 had the least 100-seed weight (77.69 ± 1.83 g). The Tukey HSD test revealed a significant difference in the 100-seed weight between the six peanut varieties *P* = 0.000 (Table 3).

#### 2.3.6. Pods per Plant

The variety with the highest number of pods per plant was CG12 (72.19 ± 3.53), followed by ICGV-SM 90704 (62.88 ± 5.73). The locally used varieties 12991 (36.12 ± 3.02) and the control (24.47 ± 1.15) had the least number of pods per plant. The Tukey HSD test revealed a significant difference between the number of pods per plant in the six peanut varieties *P* = 0.000 (Table 3).

#### 2.3.7. Shelling Percentage

A significant difference was observed in the shelling percentage among the six varieties. CG9 had the highest shelling percentage (49.21 ± 5.79%), followed by ICGV-SM 90704 (43.86 ± 4.54%), while the least was CG12 (38.62 ± 1.19%). The Tukey HSD test at 0.05 significance confirmed that there was no significant difference in the shelling percentage between the six peanut varieties *P* = 0.288 (Table 3).

### 2.4. Aflatoxin Accumulation among the Five Varieties

Variety ICGV-SM 90704 had the lowest total aflatoxin accumulation (1.82 ± 1.41 ug kg^−1^), followed by CG7 at 2.81 ± 1.99 ug kg^−1^. The locally used varieties (Homabay local) and 12991 portrayed the highest aflatoxin incidence at 7.11 ± 3.25 and 3.38 ± 0.70 ug kg^−1^, respectively. ICGV-SM 90704 had the lowest range of total aflatoxins (0.00–2.85 ug kg^−1^), followed by CG12 at 0.25–2.88 ug kg^−1^ (Table 4 and Table 5). The percentage of positive samples recovered in the aflatoxin analysis align the variation in the toxin accumulation levels in the six varieties (Appendix A).

## 3. Discussion

### 3.1. Agronomic Performance

Germination and flowering determine the productivity of peanut varieties. There was a statistical difference in the six peanut varieties; CG7 followed by ICGV-SM 90704 had the highest rates of germination. There was no positive correlation between the germination rate and the 50% and 75% flowering. However, a shorter germination period enhances the vigor of the plant and protecting it from diseases and nutritional deficiencies, which could affect its productivity [13]. Research findings revealed that peanut varieties that took least days to germinate and flower produced a higher number of pods per plant, hence were more productive [13,14]. Resistance to adverse biotic and abiotic factors associated with specific peanut strains could be an added advantage that leads to shorter germination and flowering durations. All the peanuts were grown under the same soil, environment, and agronomic practices; hence, the differing germination and flowering rates could be associated with the varietal diversity.

### 3.2. Diseases

During this study, fungal genera *Aspergillus, Cercospora,* and *Puccinia* were isolated from the diseased plants associated with causing five types of identified diseases. Foliar diseases including ELS, LLS, rust, and groundnut rosette formed 80% of all diseases while one disease (*Aspergillus* crown rot) was a stem and root disease. Eighty percent of all diseases were fungal, and 20% were viral. Fungal species are more abundant in soils, which could be the reason for the higher incidence of fungal diseases in peanuts [4]. *Aspergillus* crown rot, the disease that had the lowest incidence, prefers soils with high humidity and high temperature. The low moisture in the study region could be associated with the low incidence of *Aspergillus* crown rot due to the limited survival of its causal agent when compared to the foliar diseases. Plant leaves are rich in nutrients since they are the major sources of food for the entire plant. The higher nutrient concentration could be associated with the preference for pathogenic microbes to attack leaves, thus causing foliar diseases compared to the other plant parts. The results are supported by the findings of Ashish [7], who found that foliar diseases of peanuts formed the largest percentage of diseases. The identified disease types in the study align with findings of Sudini et al. [15], in which foliar fungal diseases were shown to be the most prevalent in peanuts.

The newly improved varieties had lower disease incidence compared to the locally used varieties. Despite the differing rates of disease infection in the newly improved varieties, they were within the same range of infestation, confirming their resistance to disease-causing pathogens. The genetic composition of the newly improved varieties combining diverse positive traits could be attributed to the higher disease resistance, unlike the locally used variety whose genetic composition is not newly improved. The findings correlate with the results of Menza et al. [16], where locally used varieties portrayed higher susceptibility to a broader range of diseases than newly improved varieties based on germplasm aspects.

### 3.3. Yield Performance

There was a significant difference in yield (pod weight, biomass, 100-pod weight, 100-seed weight, harvest index, and pods per plant) among the six peanut varieties, while no significant differences in the shelling percentage were observed between the varieties. The locally used varieties showed the lowest total pod weight, biomass, 100-pod weight, and pods per plant compared to the newly improved varieties. CG9 had the highest 100-seed weight while CG12 had the lowest, this could be attributed to the seed size differences among the varieties. The genetic composition of the newly improved varieties and their pretesting could have prevented them from biotic and abiotic barriers that compromise yield [15,16]. Newly improved varieties with enhanced germplasm and prior testing give higher yields than the locally used varieties.

There was a negative correlation between the yield parameters of all varieties and the overall disease incidence, meaning a higher rate of disease contributed to yield reduction. The high susceptibility to diseases could have resulted in reduced yield since the diseases cause defoliation and, hence, alteration of the nutritional process and protective mechanisms in the crop. Foliar diseases, which were more abundant than root and stem diseases, had a higher effect on yield reduction. Defoliation of the leaves that are a central source of nutrients for the plant could lead to lower yields. The findings are in line with those of Sudini et al. [15], who reported that approximately 80% of pod losses are caused by early and LLS, even though the percentage could change depending on the cultivar and cultural practices applied. Leaf spots produce small chlorotic lesions that later turn dark or brown and end up drying out, causing significant defoliation [16]. Losses associated with peanut rust could be 50% of the anticipated yield [7,15]. Both leaf spot and rust result in leaf necrosis and total dying, affecting the role of leaves in photosynthesis and thus compromising the yield of the crop [6]. Findings by Ashish [7] and Sudini et al. [15] confirm that foliar diseases collectively reduce the green-leaf area available for photosynthesis and stimulate leaflet abscission, leading to extensive defoliation and yield reduction.

*Aspergillus* crown rot did not have a significant effect on the yield parameters of the six peanut varieties. Very low levels of the diseases in the varieties could have caused the lack of correlation since the disease rate was too low to affect pod weight and biomass. *Aspergillus niger* greatly infects seeds; hence, it exhibits a lower rate of spread compared to foliar diseases. Unlike the foliar diseases of peanut, the stem and root disease (*Aspergillus niger*) causes approximately 10% losses in yield and is less destructive [17]. These findings correlate with findings by Matloob and Juber [18], where root diseases cause less destruction, unlike foliar diseases that cause defoliation and alter the nutritional status of the plants.

### 3.4. Aflatoxins

The locally used varieties (Homabay local or control and 12991) had the highest levels of total aflatoxins, while ICGV-SM 90704, a newly improved variety, showed the least total aflatoxin contamination. In CG7, AFB1, which is the most toxic type, had the highest incidence, followed by AFG2, which was also the most prevalent type in ICGV-SM 90704, 12991, and CG9. AFG1 had the highest prevalence in CG12. The diversity in the varieties of aflatoxin infestation could be attributed to their susceptibility to each of the four types of aflatoxins, for which some were more prone to AFG2 and AFG1 compared to the expected AFB1. However, the incidence of the four types of aflatoxins among the six varieties contradicts the findings of Er Demirhan and Demirhan [19] and Abdulrauf [20], where AFB1 was found to be the most abundant. The calibration curves for the four types of aflatoxins tested show the accumulation in the varying peanut varieties (Appendix A). Based on KEBS regulatory limits for total aflatoxins (≤10 μg kg^−1^), all the varieties were within the acceptable safety threshold for human consumption [16]. The determination of aflatoxin levels immediately after harvesting could be associated with the low levels of aflatoxin, since storage and processing are major exposures of the pods and seeds to aflatoxin-producing fungi contamination [16,19]. Before the aflatoxin analysis, the seeds had been dried to moisture levels between 7% and 10% as per the recommended standards, which further reduced the risk of high levels of the aflatoxins. The findings of Menza et al. [16] and Omara et al. [10] confirm that aflatoxins are mainly a storage problem, hence the lower levels in the peanuts tested before storage. Poor storage increases risk of aflatoxin accumulation hence peanuts could have lower levels of aflatoxin immediately after harvesting.

Locally used varieties, which were the highest diseased, had higher levels of aflatoxins compared to the least diseased newly improved varieties. The higher levels of total aflatoxins in the highest diseased varieties could be attributed to predisposing conditions to rotting and mold in plants affected by different types of diseases. The defoliation of leaves and the entire plant, caused by the foliar diseases that were the most dominant in the peanuts, compromise seed quality and make them prone to aflatoxin-causing fungal species. Menza et al. [16] and Mutegi et al. [9] corroborate the current findings in which newly improved peanut varieties with higher resistance to diseases portray lower susceptibility to aflatoxin-producing fungal species. Quality drying, careful storage, and prevention of disease infection in peanuts are essential for the prevention of aflatoxin contamination [4,8]. However, the locally used varieties, which were more susceptible to diseases, show more risk of aflatoxin contamination even after effective drying. This is opposed to the findings of Mutegi et al. [9], who found that drying of cereals reduces the risk of infestation with aflatoxin-producing fungi, and hence a lower accumulation of mycotoxins. Menza et al. [16] states that additional factors such as genetic composition, environment, and handling expose the crop to aflatoxins even after effective drying.

## 4. Conclusions and Recommendations

The results of this study show that newly improved varieties have better agronomic performance and higher resistance to diseases, which can be associated with lower aflatoxin contamination and higher yield compared to the locally used varieties. Among the newly improved varieties tested, ICGV-SM 90704 was the most resistant and showed outstanding performance, followed by CG7 and CG12, while CG9 had poorer performance. These findings can be associated with the diversity in the germplasm of each variety, which influences its vegetative performance and disease infestation characteristics. The best varieties could be adopted for cultivation in the target region and other areas with similar climatic conditions, since they would withstand the harsh environment and provide food safety and security. The locally used variety could be subjected to genetic improvement to ensure that it becomes more resistant and attains higher yield parameters.

For future research developments, it would be interesting to complement the varietal agronomic assessments on yield performance, disease incidence, and aflatoxin accumulation with a broader spectrum assessment of the impacts on the use of the different varieties in other domains contributing to food security, such as environmental, economic, social, and human aspects.

## 5. Materials and Methods

### 5.1. Field Location

The study was conducted in the semiarid agroecological zone of Nyakach subcounty, Kisumu County, Kenya, that covers an area of 182.6 km^2^ [20]. The area lies between longitude 34°44′ E and 35°15′ E and latitude 0°08′ S and 0°27′ S [11]. The altitude ranges from 1100 m above sea level (m.a.s.l) along the shores of Lake Victoria to 1800 m.a.s.l. on the Nyabondo Plateau, sharing climatic conditions with the Lake Victoria Basin plateau [12]. The region experiences annual average rainfall of 600 mm and temperatures range from 18 to 34 °C [12]. Ochola and Obuoyo [11] state that during the short rains average rainfall is 150 mm, and 700 mm during long rains.

### 5.2. Experimental Design and Treatments

The field experiment was conducted in a complete randomized block design involving six peanut (*Arachis hypogea* L.) varieties and four sites with three replications per farm. Two of the varieties, 12991 and the control (Homabay local) were obtained locally, while four of them, namely CG7, CG9, CG12, and ICGV-SM 90704, were obtained from International Crops Research Institute for the Semi-Arid Tropics (ICRISAT), Malawi (Appendix A). The region of Nyakach was selected for the growth of the peanuts since it represents one of the groundnut-producing areas in Kenya. The experiments were conducted in alignment with locally recommended cultural practices of ploughing, row planting, followed by molding, and two times of manual weeding (Table 6). Three rows of maize surrounding each experiment were planted acting as the border crops. Fifty grams of Ridomil gold (4% *w*/*w* metalaxyl-M and 64% *w*/*w* mancozeb) from Syngenta and Duduthrin 1.75EC (Twiga chemicals) contact insecticide (65 mL in 20 L of water) (Appendix A), were used in spraying the peanuts in three sessions against diseases, insects, and other pests. The control variety (Homabay local) did not receive the agronomic interventions applied to the experimental varieties and was allowed to grow under the natural setting of peanut farming in the region (Table 6).

### 5.3. Determination of Agronomic Performance

Germination: The number of seedlings that had germinated at the 6th, 9th, and 15th day from the date of planting was counted per row per variety. The number of seedlings that germinated was converted into ratios of the total number of planted seeds per row.

Flowering: The number of days required for at least 50% and 75% of the peanut plants to flower was recorded per row per variety.

### 5.4. Disease Scoring

Occurrence and severity of disease were assessed based on visual rating using the ICRISAT scale and prior prepared images and descriptions of viral, fungal, and bacterial diseases [21]. The leaves, stems, and peanut pods before were visually analyzed for symptoms that align with specific diseases. The diseased plants in each row of the four farms from germination, flowering, to harvesting were counted and recorded. The incidence was calculated by dividing the total number of infected plants with the total number of plants per row in a plot and multiplying by a hundred.

Incidence = (Total number of diseased plants/total number of germinated plants) × 100.

The presence of the visual symptoms of a disease led to the careful picking of samples of diseased plants, placing them between two newspapers and drying them with cardboard pressing for preservation purposes. The diseased parts were then transported in cooler boxes to the laboratory for further analysis and confirmation of the diseases.

### 5.5. Fungal Disease Analysis

Isolation of fungi from plants was conducted according to Taufiq et al. 2018 and Tong et al. 2011 [22,23]. Tissue samples were washed under running tap water to remove surface soil, dust, and other contaminants. Samples with overgrown saprobes were swabbed with 70% ethanol (Scharlau, S.L; Sentmenat, Spain; Appendix A). The diseased tissue pieces were cut from the leading edge of lesions using sterilized scalpels. The material was placed in 1% sodium hypochlorite for one minute. The tissue pieces were transferred from the sterilizing solution and washed by moving these briefly to sterile distilled water in three washes. The plant materials were dried on sterile filter paper, under filtered air in a laminar flow hood, and cut out into small tissue pieces (approx. 2 × 2 mm) and plated onto potato dextrose agar (PDA, Hi-media) (Appendix A). Isolation plates were incubated at 25 °C, and examined daily. Data were collected from the third day to the seventh day of culture, observing, and recording the fungal growth morphology and sporulation in terms of colors, shape, texture, and sclerotia. Pure cultures were obtained from the primary isolation plates by plating on freshly prepared PDA and incubating at 27 °C for five to seven days.

### 5.6. Identifying Fungal Isolates Using Morphological Methods

Filamentous fungi were identified at the genus level according to macroscopic and microscopic features following Pitt and Hocking [24]. Fungal isolates identified to species level were *Aspergillus spp*. and *Cercospora spp*. according to Klich [25] and Behrooz [26], respectively. The *Puccinia spp*. was identified by observing the symptoms on the leaves and microscopic features according to Hubert et al. [27].

### 5.7. Yield Parameters: Total Pod Weight, Biomass, Harvest Index, 100-Seed Weight, 100-Pod Weight, Shelling Percentage

The total pod weight was determined by weighing the total harvested pods per row per variety and expressing the weight in kg as a percentage of the total rows planted. The biomass was determined by weighing the total biomass per row per variety and expressing the weight in kg as a percentage of the total rows planted per farm. The harvest index was expressed as the pod weight of each variety as a ratio of the biomass and total pod weight and converted into a percentage. The 100-seed and 100-pod weight were determined by counting 100 seeds or pods of each variety, weighing on a balance (Sartorius; Gottingen, Germany), and expressing in kg. The shelling percentage was calculated as the weight of empty pods in kg as a percentage of the weight of pods with seeds per variety per row.

### 5.8. Aflatoxin Analysis

#### 5.8.1. Aflatoxin Sample Preparation

After harvesting, the peanut pods were dried on canvas under the sun for 14 days until the attainment of the recommended moisture content of between 7% and 10% [28]. For homogeneity, the dried pods were sampled from five points on the drying canvas holding each variety. Forty grams of seeds of each of the selected varieties (CG7, CG9, CG12, ICGV-SM 90704, 12991, control) were shelled and ground into a diameter of approximately 0.5 mm separately using a grinder. The ground seed samples were stored in airtight plastic containers in two replicates each containing 20 g and labeled in readiness for aflatoxin analysis in the mycotoxin and nutrition analysis laboratory, which is ISO/IEC 17925;2017 accredited by the Kenya Accreditation Service.

#### 5.8.2. Extraction of Aflatoxin in Peanut

A sample of 5.0 ± 0.1 g was weighed accurately into a 50 mL polypropylene centrifuge tube. A dose of 1.0 ± 0.1 g of sodium chloride salt was added to the sample. Twenty-five milliliters of 70% methanol was added into the 50 mL Falcon tube (BD, Franklin Lakes, NJ, USA) containing 5 g of milled peanut and sodium chloride. The mixture was vortexed for 1 min and shaken in a mechanical orbital shaker (New Brunswick, NJ, USA) at 250 rpm for 30 min at room temperature. The supernatant was further centrifuged at 3500× *g* rpm for 10 min. A 1:1 (*v*/*v*) dilution of the extract with 1% acetic acid was performed to obtain a final volume of 2 mL per sample. The sample was filtered through a 0.2 µm PTFE syringe filter into a HPLC vial for subsequent analysis.

#### 5.8.3. Analysis Using Ultrahigh-Performance Liquid Chromatography

Chromatographic separation was performed using a Nexera UHPLC system (Shimadzu Corporation, Kyoto, Japan) fitted with a SIL-30AC auto sampler, LC-20AD Prominence pumps, and RF-20AXS Prominence Fluorescence detector (Appendix A). A Synergi Hydro-RP analytical column (2.5 µm particle size, 100 mm × 3.00 mm) (Phenomenex, Torrance, CA, USA) operating at flow rate of 0.4 mL min^−1^ was used for the separation of aflatoxins. A binary mobile phase, consisting of mobile phase A methanol (40%) and mobile phase B 1% acetic acid (60%), was utilized to achieve this separation. The injection volume was 10 µL and the column oven temperature was set at 50 °C. The liquid chromatography program was set at 8 min per run and 60% methanol was used as the flushing solution for the column [29]. Fluorescence detection was carried out at wavelengths of λ_ex_ = 365 nm and λ_em_ = 435 nm. A standard calibration curve from a plot of peak areas against the known concentration of the injected series of standards was established and used for estimating the concentrations of the samples in the LabSolutions software version 5.89 (Shimadzu Corporation, Kyoto, Japan, 2014) (Appendix A). Individual types of aflatoxin were identified by comparing the retention time of the chromatographic peak of the target aflatoxin in the experimental sample and that of the corresponding standard chromatographic peak (Appendix A). Samples with values above the linear range of the standard curve were diluted and retested. The accuracy of the reported aflatoxin was within the acceptable range at the Z score ± 2 [30]. The accuracy was confirmed by using the assigned and measured values of the control samples or materials (Appendix A). A proficiency test findings shown in the Appendix A further support the efficiency and accuracy of the procedure. Method validation was evaluated using linearity, recovery, accuracy, precision, or limit of detection (LOD) and limit of quantification (LOQ) (Table 7). Further details on the linearity from spike recovery, LOQ, and, LOD for AFB1, AFB2, AFG1, AFG2 and total aflatoxins are in Appendix A respectively.

### 5.9. Statistical Analysis

Analysis of variance (ANOVA) was used in determining the differences in means between the different peanut varieties for the agronomic characteristics, yield parameters, diseases, and aflatoxin levels. The Tukey HSD test was used in determining significant differences between means set at a confidence interval of 0.05. A Pearson correlation test was used to determine the association between diseases, agronomic characteristics, and yield of peanuts among the six different varieties. The R statistics program was applied in the data analysis.

All Appendix A could be seen in Appendix A.

## Figures and Tables

**Figure 1 toxins-15-00111-f001:**
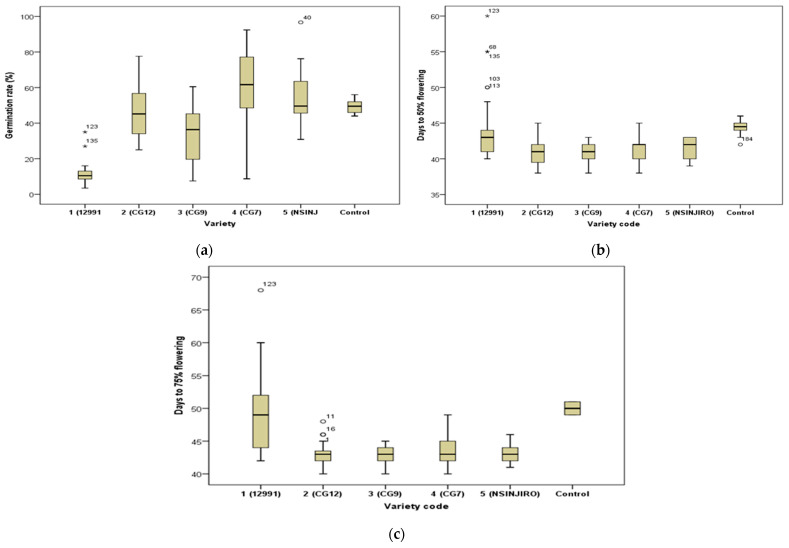
(**a**) Germination rates among the six varieties; (**b**) days to 50% flowering; (**c**) days to 75% flowering. *,^o^: Outliers.

**Table 1 toxins-15-00111-t001:** Peanut disease prevalence per variety.

Varieties	Disease Incidence (Mean ± SD) (%)					
	Rust	ELS	LLS	Groundnut Rosette Virus	*Aspergillus* Crown Rot	Mean Disease Incidence per Variety
12991	1.78 ± 0.62 ^ab^	4.79 ± 0.67 ^c^	4.29 ± 1.88 ^b^	8.29 ± 1.61 ^c^	1.34 ± 0.40 ^a^	5.41 ± 8.31
CG7	0.52 ± 0.18 ^a^	1.78 ± 0.17 ^a^	1.21 ± 0.12 ^a^	1.90 ± 0.21 ^a^	0.17 ± 0.07 ^a^	1.53 ± 2.13
CG9	0.44 ± 0.17 ^a^	3.15 ± 0.50 ^abc^	2.03 ± 0.37 ^ab^	3.7 ± 0.39 ^ab^	0.31 ± 0.13 ^a^	2.60 ± 4.17
CG12	0.69 ± 0.02 ^a^	2.08 ± 0.25 ^ab^	1.23 ± 0.24 ^a^	2.30 ± 0.33 ^a^	0.38 ± 0.12 ^a^	1.85 ± 2.74
(ICGV-SM 90704)	0.22 ± 0.16 ^a^	1.68 ± 0.33 ^a^	0.86 ± 0.21 ^a^	1.62 ± 0.38 ^a^	0.34 ± 0.16 ^a^	1.31 ± 1.75
Control	2.99 ± 0.48 ^b^	4.13 ± 0.78 ^bc^	8.92 ± 1.10 ^c^	7.64 ± 1.77 ^bc^	9.21 ± 2.50 ^b^	7.41 ± 1.88
All Varieties	0.88 ± 0.15	2.79 ± 0.20	2.34 ± 0.24	4.01 ± 0.44	0.75 ± 0.21	

a,b,c are letters showing significant differences between values. Values in the columns followed by the same letter are not significantly different at *P* < 0.05.

**Table 2 toxins-15-00111-t002:** The total pod weight and biomass of the six peanut varieties.

Variety Code	Total Average Pod Weight per Row (kg) (Mean ± SD)	Average Biomass per Row (kg) (Mean ± SD)	Average Harvest Index per Row (%) (Mean ± SD)
12991	1.60 ± 0.28 ^c^	5.45 ± 0.54 ^c^	22.00 ± 0.16 ^c^
CG12	7.29 ± 0.56 ^a^	17.67 ± 2.63 ^ab^	25.31 ± 1.22 ^ab^
CG9	4.50 ± 0.39 ^b^	14.34 ± 1.17 ^bc^	23.89 ± 1.12 ^b^
CG7	8.86 ± 1.45 ^a^	25.06 ± 2.47 ^ab^	26.12 ± 0.91 ^a^
ICGV-SM 90704	9.22 ± 1.19 ^a^	27.21 ± 5.05 ^a^	29.21 ± 0.72 ^a^
Control	1.50 ± 1.11 ^c^	6.60 ± 1.20 ^c^	18.52 ± 1.44 ^c^

a,b,c are letters showing significant differences between values. Values in the columns followed by the same letter are not significantly different at *P* < 0.05.

**Table 3 toxins-15-00111-t003:** Yield parameters of the six peanut varieties (100-seed weight, 100-pod weight, pods per plant, shelling percentage).

Yield Parameters (Mean ± SD)				
Variety Code	100-Seed Weight (g)	100 Pod-Weight (g)	Pods per Plant (Pods Plant^−1^)	Shelling Percentage (%)
12991	78.71 ± 3.17 ^a^	181.46 ± 6.44 ^bc^	36.10 ± 3.02 ^bc^	43.66 ± 1.33 ^a^
CG12	77.69 ± 1.83 ^a^	205.25 ± 5.80 ^b^	72.19 ± 3.53 ^a^	38.62 ± 1.19 ^a^
CG9	118.55 ± 6.03 ^b^	259.37 ± 6.40 ^a^	53.19 ± 2.76 ^b^	49.21 ± 5.79 ^a^
CG7	111.43 ± 2.12 ^b^	260.84 ± 4.60 ^a^	57.60 ± 1.94 ^ab^	43.28 ± 0.96 ^a^
(ICGV-SM 90704)	115.71 ± 15.24 ^b^	262.93 ± 10.8 ^a^	62.88 ± 5.73 ^a^	43.86 ± 4.54 ^a^
Control	84.40 ± 4.52 ^a^	150.80 ± 11.4 ^c^	24.47 ± 1.15 ^d^	43.60 ± 1.16 ^a^

a,b,c are letters showing significant differences between values. Values in the columns followed by the same alphabetical letter are not significantly different at *P* < 0.05.

**Table 4 toxins-15-00111-t004:** Mean of aflatoxin levels (total, AFB1, AFB2, AFG1, AFG2) in the six peanut varieties at harvest.

Variety	Mean ± SD (ug kg^−1^)				
	Total Aflatoxins	AFB1	AFB2	AFG1	AFG2
12991	3.38 ± 0.70 ^a^	0.46 ± 0.65 ^a^	0.31 ± 0.43 ^a^	1.15 ± 0.11 ^a^	1.46 ± 0.27 ^a^
CG7	2.81 ± 1.99 ^b^	0.83 ± 0.97 ^a^	0.00 ± 0.00 ^b^	0.85 ± 1.01 ^a^	1.12 ± 0.55 ^a^
CG9	2.96 ± 3.26 ^b^	0.00 ± 0.00 ^a^	0.00 ± 0.00 ^b^	2.25 ± 3.62 ^b^	0.72 ± 0.85 ^b^
CG12	3.36 ± 3.68 ^a^	0.80 ± 0.92 ^a^	0.21 ± 0.42 ^b^	1.19 ± 1.11 ^a^	0.16 ± 1.56 ^b^
ICGV-SM 90704	1.82 ± 1.41 ^b^	0.31 ± 0.62 ^a^	0.00 ± 0.00 ^b^	0.50 ± 0.57 ^a^	1.01 ± 0.43 ^a^
Control	7.11 ± 3.25 ^c^	0.00 ± 0.00 ^a^	0.56 ± 0.28 ^a^	6.29 ± 3.16 ^c^	0.27 ± 0.37 ^b^

UHPLC-FLD was used for aflatoxin analysis; numbers are replication of three tests. AFB1: Aflatoxin B1, AFB2: Aflatoxin B2, AFG1: Aflatoxin G1, AFG2: Aflatoxin G2. a,b,c are letters showing significant differences between values. Values in the columns followed by the same alphabetical letter are not significantly different at *P* < 0.05.

**Table 5 toxins-15-00111-t005:** Range of aflatoxins (total, AFB1, AFB2, AFG1, AFG2) in the six peanut varieties at Harvest.

Variety			Range of Aflatoxins (ug kg^−1^)		
	Total Aflatoxins	AFB1	AFB2	AFG1	AFG2
12991	2.88–3.87	0.00–0.9	0.00–0.61	1.07–1.23	1.27–1.65
CG7	0.61–5.38	0.00–1.82	0.00–0.00	0.00–2.01	0.61–1.86
CG9	0.00–7.58	0.00–0.00	0.00–0.00	0.00–7.58	0.00–1.69
CG12	0.25–2.88	0.00–1.70	0.00–0.84	0.00–2.62	0.00–3.46
ICGV–SM 90704	0.00–2.85	0.00–0.1.24	0.00–0.00	0.00–1.01	0.67–1.6
Control	4.81–9.41	0.00–0.00	0.36–0.76	4.05–8.52	0.00–0.53

**Table 6 toxins-15-00111-t006:** Agricultural practices adopted for the five peanuts varieties and the control at the four sites.

Agricultural Practices	Varieties	
	12991, CG7, CG9, CG12, ICGV-SM 90704	Control
Ploughing		
Dates (2021)	5–12 February	5–12 February
	24–26 February	24–26 February
Planting	27 March	
Space inter-row (m)	0.50	Broadcasted
Space between plants within row (m)	0.10	Broadcasted
Manure application		
Dates (2021)	17–19 February	No manure
Rates (t ha^−1^)	14,830	No manure
Fungicide application		
Specifications	Ridomil	None
No. of applications	3	None
Days after planting	45, 59, and 73	None
Rates (g/L of water/plot)	50/20/plot	None
Insecticide application		
Specifications	Duduthrin	None
N. of applications	3	None
Days after planting	45, 59, and 73	None
Rates (mL/L of water/plot)	65/20/plot	None
Molding		
N. of interventions	1	1
Growth stage	Flowering	Flowering
Manual weeding		
N. of interventions	2	1
Growth stage	Germination, flowering	Flowering

**Table 7 toxins-15-00111-t007:** Linearity, spike recovery, limits of detection, and limits of quantification.

Toxin	Calibration Standards Linearity	Spike Recovery						
	Range	Level 1 7.5 ng/g	Level 2 15 ng/g	Level 3 30 ng/g	Average Spike Recovery	Spike Recovery Standard Deviation	LOD (ug kg^−1^)	LOQ (ug kg^−1^)
AFB1	0.52–104.1	77.6	74.01	74.43	75.36	1.63	0.79	2.64
AFB2	0.52–103.9	74.06	75.65	74.14	74.62	0.73	0.56	1.86
AFG1	0.52–103.9	102.32	84.18	76.52	87.67	10.82	0.26	0.88
AFG2	0.52–104.1	58.16	75.20	80.92	71.42	9.67	0.40	1.32
Total aflatoxins							1.03	3.42

## Data Availability

Data is contained within the article or Appendix A.

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
