# Peer review of "Evaluation of Agronomic Characteristics, Disease Incidence, Yield Performance, and Aflatoxin Accumulation among Six Peanut Varieties (Arachis hypogea L.) Grown in Kenya"

_toxins, 2023, doi:10.3390/toxins15020111_

Round 1

Reviewer 1 Report

The manuscript entitled “Evaluation of Agronomic Characteristics, Disease Incidence, Yield Performance, and Aflatoxin Accumulation among Six peanut Varieties (Arachis hypogea L.) Grown in Kenya” described interesting research about the differences between improved varieties of peanuts with the local ones. These results could contribute to reduce the production losses and the risk associated to aflatoxins intake. The manuscript is well-organised, and if the authors address the following issues, it should be considered for publication.

Lines 15 and 18: the disease incidence is a percentage. 100-pod weigt is in kg? Add units.

check this throughout the manuscript.

Line 44: specify which are the diseases.

Line 63: after 1.0 add the unit

Line 70: The toxic and carcinogenic nature of the mycotoxins…

Line 86: germination rate, unit?

Figure 1: It is needed to improved the figure quality. The words are not clear.

Table 4: no differences between 12911 in total aflatoxins and 6,27 from the control sample?? Differences between CG9 in AFB1 and Control when both have 0 ug/kg? Revise the statistical analysis.

Table 5: how is it possible to have a range with a maxium of 1.37 ug/kg for AFB1 in CG7 and in the previous table the deviation is higher than the maximum level (1.69)?? Revise the data.

Line 288: substitute microbes by microorganisms

Line 318 to 323: This paragraph seems part of the introduction rather than material and methods

Line 335: Which are the trademark from Ridomil and Duduthrin and their main components?

Table 6: The table must not repeat the information that it is written in the text, is redundant and unnecessary. This table could be smaller without repetitions and grouping the varieties with the same treatments in the same column.

Line 369: if you wash the samples under running tap water you are altering the fungal population and more if you then sterilise the samples with 1% sodium  hypochlorite for one minute and afterwards in 10% ethanol. Are you sure that part of the fungal population is not eliminated in this way?

Line 384: specify more details about the fungal identification. The macroscopic and microscopic evaluation only allows to identify the to gender level, not species.

Line 385: How did you identify Cercospora if it is not mentioned in Klich 2007. Add references for Puccinia spp.

Line 428: Did you used a clean up column as it is indicated in this reference from AOAC?

Line 437: Add the limits of detection and quantification of the HPLC-FLD method.

Reviewer 2 Report

Dear Authors,

There are  some comments, questions and suggestions regarding to the paper of Yours:

General

The article lacks a section with a description of the reagents and materials used in the research. In addition, sometimes there are no producers of used equipment. This should be supplemented. The numbers of significant digits in results values should be unified (tables 3,4,5). The authors could supplement the publication with a section with abbreviations. Abbreviations should be avoided in the abstract.

Line 18: unit is missing (100-pod weight)

Figure 1: the quality of the figures should be improved

Table 4: the statistic in the table is probably not correct, the authors should check it. E.g. AFB1- for CG9 0±0a, while for control 0.00±0.0c

Table 4 and 5: in some cases the data do not match, e.g. table 4 mean total aflatoxins for ICGV is 0,49 ug/kg while in table 5 range for total aflatoxin is from 0,85 to 1,95 ug/kg. Authors should verify that the data in these tables are correct.

Table 5: could the authors supplement Table 5 with the percentage of positive samples for the given analytes

Table 6: whether this table is needed if everything is described in paragraph 5.2 (except data for control samples that could be added)

Line 382: regarding to microbiological analysis, could the authors supplement the data with the quantitative assessment of fungi expressed as the colony-forming unit (cfu) per gram of peanut samples

Line 398: In aflatoxins analysis there were no advanced clean-up as well as neither pre- or post-column derivatization applied. Taking into account relatively low aflatoxins determined, a question about the limits of detection ad quantification arises. Could the authors supplement the article with a description of the validation of the analytical method and provide the values of the parameters obtained during validation procedure  (recoveries, precision, method range, LOD and LOQ values).

Round 2

Reviewer 1 Report

The quality of the manuscript has been improved after the suggestions and it can be published now. 

Reviewer 2 Report

Dear Authors,

thank you for considering my comments and correcting the manuscript.

I would also like to ask you to add (to the supplementary data file) 2 aflatoxins chromatograms:

-         1)  Peanuts sample containing aflatoxins

-        2)   calibration curve standards sample (with a similar concentration of analytes like in 1 chromatogram)
